# FORMAL MATHEMATICS STATEMENT CURRICULUM LEARNING

**Stanislas Polu**
OpenAI

**Jesse Michael Han**[†]
Multi Technologies

**Kunhao Zheng**
École Polytechnique

**Mantas Baksys**
University of Cambridge

**Igor Babuschkin**[†]
DeepMind

**Ilya Sutskever**
OpenAI

## ABSTRACT

We explore the use of expert iteration in the context of language modeling applied to formal mathematics. We show that at same compute budget, expert iteration, by which we mean proof search interleaved with learning, dramatically outperforms proof search only. We also observe that when applied to a collection of formal statements of sufficiently varied difficulty, expert iteration is capable of finding and solving a curriculum of increasingly difficult problems, without the need for associated ground-truth proofs. Finally, by applying this expert iteration to a manually curated set of problem statements, we surpass previous state-of-the-art on the *miniF2F* benchmark, automatically solving multiple challenging problems drawn from high school olympiads.

## 1 INTRODUCTION

Deep learning has enjoyed spectacular success in many domains, including language (Brown et al., 2020; Devlin et al., 2019; Wu et al., 2016), vision (Radford et al., 2021; Tan & Le, 2019), and image generation (Ramesh et al., 2021; Karras et al., 2019). One domain where deep learning has not yet enjoyed a comparable success is in tasks that require extensive *planning* and *symbolic reasoning*, with the exception of two-player games (Silver et al., 2016; 2017; Berner et al., 2019; Vinyals et al., 2019). In such games, deep learning systems exhibit a considerable degree of reasoning, especially when trained with self-play combined with a search procedure such as Monte Carlo Tree Search (MCTS) (Browne et al., 2012). But the resulting reasoning abilities achieved are limited due to the relatively narrow scope of games.

As such, theorem proving in interactive proof assistants, or formal mathematics, appears as an interesting game-like domain to tackle due to its increased scope. The typical tasks consist of generating a machine-checkable proof given a formal statements. Like games, formal mathematics has an automated way of determining whether a trajectory (*i.e.* a proof) is successful (*i.e.* formally correct). But the vast scope of formal mathematics means that any strong reasoning result obtained in it will be more meaningful than comparable results in games (*e.g.* finding proofs to mathematical conjectures), and could even be applicable to important practical problems (*e.g.* software verification).

However, tackling formal mathematics involves two main challenges that we must address in order to continue making progress:

**Infinite action space** Not only does formal mathematics have an extremely large search space (like Go (Silver et al., 2016) for example), it also has an infinite action space. At each step of proof search, the model must choose not from a well-behaved finite set of actions, but a complex and infinite set of tactics, potentially involving exogenous mathematical terms that have to be generated (e.g., generating a mathematical statement to be used as a witness, an object used steps such as "there exists an $x$ ...", or a cut, the introduction and the chaining of a lemma in the middle of a proof).

**No direct self-play setup** In formal mathematics, a prover is not playing against an opponent but against a set of statements to prove. When faced with a statement that is just too hard, there is no

---

[†]Work performed while at OpenAI.

obvious reframing of the formal mathematics setup that will let the prover generate intermediary easier statements to tackle first. This asymmetry prevents naive application of the symmetric self-play algorithms commonly used in 2-player games.

These two differences make a naive application of reinforcement learning to formal mathematics leave a large room for improvement (Whalen, 2016; Winands et al., 2008). Past work proposed to address the infinite action space problem by sampling from a language model (Polu & Sutskever, 2020), while training such language model requires a large dataset of statements with proof. This paper focuses on this second problem and our basis for addressing it is the observation that the key role of self-play is to provide an unsupervised curriculum. We propose instead to supply auxiliary sets of problem statements (without requiring proofs) of varying difficulty. We empirically show that, when the difficulty of these auxiliary problems is varied enough, a simple expert iteration procedure is able to solve a curriculum of increasingly difficult problems, eventually generalizing to our target distribution. We show that this works with both automatically-generated and manually-curated auxiliary distributions of problems and leverage this to achieve state-of-the-art on the *miniF2F* benchmark. Our results suggest that continuous self-improvement in formal mathematics can potentially be reduced to the problem of generating such sets of formal statements, which we have done in part manually in this work, but could eventually be scaled in the future with more automation (such as more domain-specific statements generator or even informal to formal machine translation).

***miniF2F* benchmark** In this work, we target the *miniF2F* (Zheng et al., 2022) benchmark, which consists of 244 *validation* and 244 *test* formalized statements of mathematical problems from various competitions. We believe it to be a better measure of mathematical reasoning compared to a formal library-derived split. Also, the extreme scarcity in formal libraries of this type of problems makes it an ideal test-bed for the expert iteration methodology studied in this paper.

## 2  RELATED WORK

Our work strongly relies on, and can be seen as a natural continuation of the work presented in the original GPT-f paper (Polu & Sutskever, 2020) which studies the use of language models to generate tactics, the PACT paper (Han et al., 2022) which applies GPT-f to Lean and studies the benefits from co-training on self-supervised objectives, and the *miniF2F* benchmark (Zheng et al., 2022). We present additional related work in Appendix A.

## 3  FORMAL ENVIRONMENT

We choose Lean (de Moura et al., 2015; lea) as our formal environment. Unlike Metamath (Megill & Wheeler, 2019) , which has been studied in the original GPT-f paper (Polu & Sutskever, 2020), Lean benefits from high-level tactics which were shown to be beneficial in the context of the *miniF2F* benchmark. Also, Lean has recently received a lot of attention from the mathematical community, thanks to projects such as the Perfectoid Spaces (Buzzard et al., 2019) and the Liquid Tensor experiment (Scholze, 2020), and benefits from a vibrant community of hundreds of contributors to its main mathematical library called *mathlib*. We refer to the PACT paper's Background section (Han et al., 2022) for a detailed introduction to Lean in the context of neural theorem proving. We refer to Appendix D for an illustration of *miniF2F* input and Lean environment.

**`lean-gym`** In the PACT paper (Han et al., 2022), proof search is performed by the Lean runtime using the LEANSTEP environment, with a generic backend interface to models. While easy to use–one just needs to plug in their model–this approach makes it difficult to alter and iterate on the search procedure because it is programmed in Lean (which is not designed or intended for cluster-wide parallelised I/O intensive tasks), and the coupling of the search procedure with the Lean runtime introduces challenges when scaling to a large number of parallel workers.

To solve these issues we implemented `lean-gym`[1] – a simple REPL interface over the standard input/output implemented in Lean directly. We present `lean-gym`'s API and discuss some of its advantages and limitations in Appendix B.

---

[1] https://github.com/openai/lean-gym

**Proof extraction** We rely on the proof extraction methodology presented in the PACT paper (Han et al., 2022) to extract human tactic proof steps from *mathlib* (the `tactic` dataset) as well as the various other proof artifacts (`mix1` and `mix2` datasets). We also extract *mathlib-{train, valid, test}*, the set of statements from *mathlib* along the split proposed in Han et al. (2022) (the *validation* and *test* splits of `tactic, mix1, mix2` being aligned with *mathlib-{valid, test}* as the splits are determined by declaration name hashes (across all data sources including proof-term mining) as opposed to individual proof steps or data-points.

## 4 EXPERT ITERATION

*Expert iteration* was introduced in Silver et al. (2017) and broadly consists in iteratively training models on their previously sampled trajectories, to achieve continuous improvement. In this section we present our expert iteration methodology, including the models and pre-training strategies. We use decoder-only Transformers similar to GPT-3 (Brown et al., 2020). Throughout this paper we focus on a model with 36 layers and 774 million trainable parameters (referred to as the *700m* model in the GPT-f paper (Polu & Sutskever, 2020)).

### 4.1 PRE-TRAINING

We pre-train our models successively on GPT-3's post-processed version of CommonCrawl (for 300B tokens) and an updated version of *WebMath* (Polu & Sutskever, 2020) (for 72B tokens) whose mix is presented in Appendix C.

### 4.2 TRAINING OBJECTIVES

***Proofstep objective*** The *proofstep objective*, introduced in Polu & Sutskever (2020), consists in generating a PROOFSTEP (a Lean tactic) given a GOAL (a Lean tactic state). We also condition this objective on the current DECLARATION (a Lean theorem name), which remains the same throughout a proof search: DECL <DECLARATION> GOAL <GOAL> PROOFSTEP <PROOFSTEP>.

The rationale for conditioning on the declaration name is to hint our models on the position of the current declaration in the *mathlib* library. It can be considered as a weak proxy signal for the large amount of information not shown to the model (the full environment consisting of the available imports and currently open declarations such as module names, notations, declared instances, ...). The declaration name lets models at least in principle memorize and then retrieve some of that information, knowing that `lean-gym` errors if a theorem or definition that is not available in the environment associated with the current declaration is used by tactics generated by our models. Also note that conversely to Polu & Sutskever (2020) and like Han et al. (2022) <GOAL> is not necessarily a single goal but a Lean tactic state, which possibly comprises multiple goals.

***Proofsize objective*** We depart from Polu & Sutskever (2020) and use a *proofsize objective* to guide our proof searches, which consists in generating one token that represents a proof size estimate bucket for the current goal (Lean tactic state): DECL <DECLARATION> GOAL <GOAL> PROOFSIZE <PROOFSIZE_BUCKET_TOKEN>

For a given goal $g$, either the goal was proved as part of the proof search and we denote its proof size (the number of tactic applications (compounded Lean tactics counting as one)) as $ps(g)$, or the goal was not proved in which case we assign the goal to a bucket that virtually represents "infinite" proof sizes.

We use 11 buckets $B = 0...10$ and compute the *proofsize* bucket $b(g)$ for a goal $g$ by assigning infinite proof sizes to bucket 0, all proof sizes over 20 to bucket 1 and linearly projecting proof sizes lower than 20 on the remaining buckets 2, ..., 10 (10 being the bucket for the shortest proof sizes). In practice, when training and sampling from the model, we map $B$ to the tokens 'A'...'K'.

To value goals as we run proof searches, we sample the *proofsize* bucket token and record the probability $p_b(g)$ for each viable bucket and use them to get a weighted average with the following formula: $v(g) = \frac{1}{\#B} \sum_{b \in B} p_b(g) \cdot b$. As an example, if the model assigns $p_0 = 1$ (hence $p_{b \neq 0} = 0$) then $v(g) = 0$. Conversely if the model assigns $p_{10} = 1$ (10 being the bucket for the shortest proof sizes) then $v(g) = 1$.

Table 1: Performance of $\theta_0$ and $\theta_1$ on *mathlib-valid* and *miniF2F-valid* compared to PACT Lean GPT-f as reported in Han et al. (2022); Zheng et al. (2022). All models have the same architecture. $\theta_0$ is sampled using cumulative logprob priority best-first search. $\theta_1$ is sampled using best-first search based on the *proofsize objective*. We report our setup ($d = 512$ expansions and $e = 8$ tactic samples per expansions) as well as the setups used in Han et al. (2022); Zheng et al. (2022) (denoted as $\theta_0^*$) to control for compute. We also report the performance of $\theta_1$ on *mathlib-valid* when trained using the *outcome objective* (denoted as $\theta_1'$) from Polu & Sutskever (2020) as an ablation of our proposed *proofsize objective*.

| Model | $d$ | $e$ | *pass@1* | *pass@8* | Model | $d$ | $e$ | *pass@1* | *pass@8* |
|---|---|---|---|---|---|---|---|---|---|
| *mathlib-valid* | | | | | *miniF2F-valid* | | | | |
| PACT | 512 | 16 | 48.4% | | miniF2F | 128 | 16 | 23.9% | 29.3% |
| $\theta_0^*$ | 512 | 16 | 48.5% | 57.6% | $\theta_0^*$ | 128 | 16 | 27.6% | 31.8% |
| $\theta_0$ | 512 | 8 | 46.7% | 57.5% | $\theta_0$ | 512 | 8 | 28.4% | 33.6% |
| $\theta_1$ | 512 | 8 | **56.3%** | **66.3%** | $\theta_1$ | 512 | 8 | **28.5%** | **35.5%** |
| $\theta_1'$ | 512 | 8 | 55.6% | 65.9% | $\theta_1'$ | 512 | 8 | 28.3% | 34.7% |

The rationale for using this *proofsize objective* instead of the *outcome objective* described in Polu & Sutskever (2020) is that (i) it achieves better performance compared to the *outcome objective* (see Table 1), and (ii) it prioritizes goals that potentially lead to shorter proofs during proof search, creating an intrinsic incentive for the system to converge towards shorter proofs. Similarly to Polu & Sutskever (2020) we favor this token-based approach to the introduction of a separate value head to keep the overall architecture simple. This way the *proofsize objective* can be implemented by simply augmenting the training dataset and without any architectural change.

### 4.3 BOOTSTRAPPING

Bootstrapping consists in the steps required to train an initial model on both the *proofstep objective* and the *proofsize objective*.

Given a pre-trained model on *WebMath*, we fine-tune it on the `tactic` dataset extracted from *mathlib* as well as the proof artifacts dataset `mix1` as described in Han et al. (2022). This initial model, which we denote $\theta_0$ is solely trained on the *proofstep objective*. We use the *validation* splits of the `tactic` and `m1` datasets to early-stop training. Note that this is our only use of *mathlib-valid* to influence the training process throughout this paper.

To generate data for the *proofsize objective*, we use $\theta_0$ to sample proofs for statements from *mathlib-train*. For each statement from *mathlib-train* (25k) we attempt $a = 1$ proof searches using the cumulative logprob priority search described in Polu & Sutskever (2020) (which does not require a trained value function) using $d = 512$ expansions and $e = 8$ samples per expansion. We denote the set of successful proof searches created in this process as $S_0$.

Using $S_0$ we generate dataset $D_0$ by concatenating: (i) the initial `tactic` dataset (*proofstep objective*), (ii) a deduplicated set of proofsteps extracted from the proofs in $S_0$ (*proofstep objective*) and (iii) a deduplicated set of proofsize tuples (goals and proofsize) extracted from the full proof searches in $S_0$ (*proofsize objective*).

Note that the full proof searches in $S_0$ include goals that are visited but eventually remain unproved, which provides useful negative examples for the trained value function (even if these negatives may include provable goals that simply were not prioritized by the search). Also note that $S_0$ doesn't include failed proof searches.

We fine-tune $\theta_0$ on $D_0$ for exactly one epoch (no use of *validation* data for early-stopping) to obtain our initial model $\theta_1$ trained on both the *proofstep objective* and the *proofsize objective*. $\theta_0$ is used in our expert iteration setup as base model to fine-tune from at each iteration, and $\theta_1$ is our first iterated model or *mathlib* bootstrapped model trained on both objectives.

We report in Table 1 the pass rates of $\theta_0$ and $\theta_1$ on *mathlib-valid* and *miniF2F-valid* and compare with previously reported pass rates for equivalent amounts of compute. As reported in Polu & Sutskever (2020), training a value function to guide search greatly improves the pass rates of $\theta_1$ on *mathlib-valid*.

Interestingly, the gap between $\theta_0$ and $\theta_1$ on *miniF2F-valid* is not as significant, demonstrating that training a value function on proofs sampled from *mathlib-train* has limited transfer to *miniF2F-valid*. The main differences with Zheng et al. (2022), potentially explaining the gap on *miniF2F-valid* ($27.6\%$ vs $23.9\%$), consists in the new pre-training described in Section 4.1 as well as the use of a more recent *mathlib* checkpoint for the `mix1`, `mix2` and `tactic` datasets.

## 4.4 ITERATED SAMPLING AND TRAINING

Our expert iteration process takes as input: (i) a set of formal statements $St$, (ii) a function $a : St \to \mathbb{N}$ indicating the number of proof search attempts to run per statement at each iteration, (iii) a base model $\theta_0$ to fine-tune from at each iteration, and (iv) a *mathlib* bootstrapped model $\theta_1$ trained on both objectives. A high-level illustration of the iterated sampling and training is available in Appendix E.

Each iteration $k$ consists in sampling proof searches for statements in $St$ using $\theta_k$, filtering successful proof searches $S_k$ to extract a new dataset $D_k$, and fine-tuning $\theta_0$ on it to obtain $\theta_{k+1}$, on which we can iterate. To sample proof searches from $St$ we use the best-first search described in Polu & Sutskever (2020) with the value function described in Section 4.2. We attempt $a$ proof searches for each statement $s(s \in St)$ with $d = 512$ expansions and $e = 8$ samples per expansion. We denote the set of successful proof searches for iteration $k$ as $S_k$.

Using $S_k$ we generate datasets $D_k$ by concatenating: (i) the initial `tactic` dataset (*proofstep objective*), (ii) a deduplicated set of proofsteps extracted from the proofs in $\bigcup_{1 \le i \le k} S_k$ (*proofstep objective*), and (iii) a deduplicated set of proofsize tuples (goals and proofsize) extracted from the full proof searches in $\bigcup_{1 \le i \le k} S_k$ (*proofsize objective*).

We use a global deduplication across iterations for both proofsteps and proofsize tuples which we found to be important to maintain the stability of the expert iteration procedure. This global deduplication is somewhat equivalent for each statement to growing a unique proof tree by aggregating all the proof searches that have been run for it across iterations. This virtual proof tree accumulates a growing number of positive proof paths and visited goals that remain unproven. We use these goals as negative examples for the *proofsize objective*, labeling them with an infinite proofsize. Positive goals are deduplicated keeping the minimum proof sizes across proof searches.

Finally $\theta_k$ is obtained by fine-tuning $\theta_0$ for exactly one epoch on $D_k$. Note that the initial `tactic` dataset is included in each $D_k$, despite $\theta_0$ being already trained on it (along with `mix1`). We found this repetition to be beneficial overall (as it adds the *mathlib* extracted proofsteps to our deduplicated per statements virtual proof trees) despite it leading to a slight overfit on the `tactic` dataset in terms of validation loss.

## 4.5 EXPERT ITERATION ON *mathlib-train*

In this section we propose to set $St$ to the statements in *mathlib-train*, run our expert iteration process with it and report performance on both *mathlib-valid* and *miniF2F-valid*. Performance is reported in terms of pass rate (percentage of successful proof searches) as a function of the number of attempts per statement, noted $pass@k$ where $k$ is the number of attempts per statement at test time. To reduce noise in these metrics we run more than $k$ attempts at test time (generally 32 to compute $pass@1$ and $pass@8$), averaging across attempts as needed to obtain a smoother $pass@k$ value.

Given the large number of statements in *mathlib-train* (25k) we uniformly set $a = 1$ and use $\theta_0$ and $\theta_1$ as described in Section 4.3 and report *pass@1* and *pass@8* across 8 iterations in Figure 1. The *pass@1* on *mathlib-valid* goes from $56.3\%$ for $\theta_1$ to $62.6\%$ for $\theta_9$. The performance steadily improves and follows a clear logarithmic scaling law on *mathlib-valid*. It is also notable that, initially, transfer to out-of-distribution *miniF2F-valid* appears limited but eventually kicks in as we reach better performance on *mathlib-valid*. This demonstrates that the expert iteration process does not just overfit to *mathlib* but also leads to improved performance on out-of-distribution statements.

We define the cumulative pass rate at iteration $k$ as the pass rate consisting of all proof searches up to iteration $k$ . Since we set $a = 16$ for evaluation on *mathlib-valid* and *miniF2F-valid* at each iteration, the cumulative pass rate at iteration $k$ can be seen as a noisy ensembled *pass@16k* (multiple models ($\theta_k$), no averaging). In Figure 2, we report this cumulative pass rate for two iteration loops, our normal one and a sampling-only loop where we skip re-training the model between iterations

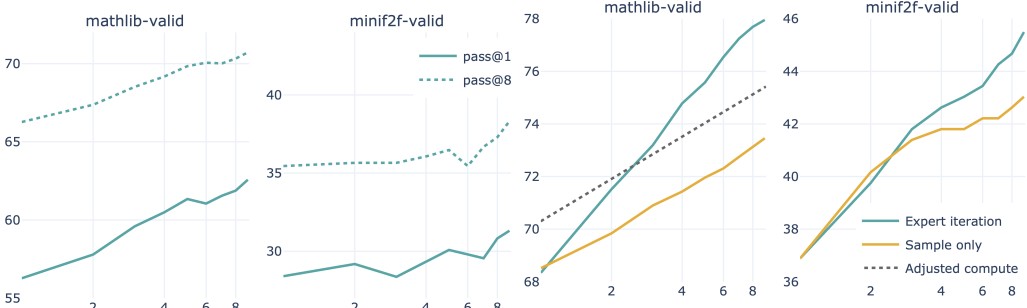

Figure 1: *pass@1* (plain) and *pass@8* (dotted) for *mathlib-valid* and *miniF2F-valid* when running 8 expert iterations with $St$ set to be the statements in *mathlib-train*. The x-axis is log-scaled. It corresponds to the indices of the $\theta_k$ models and serves as a good proxy to compute (the amount of test-time and train-time compute per iteration being fixed). The y-axis is scaled linearly and simply shifted between the two graphs (spans an equal range).

Figure 2: Cumulative pass rate for our *expert iteration* loop as well as a *sample only* loop where we skip re-training the model between iterations. The *adjusted compute* line is computed by fitting the *sample only* curve and shifting it to approximate a setup where we would focus all the additional compute used by expert iteration (sampling training data from *mathlib-train* as well as re-training models at each iteration) towards running proof searches against *mathlib-valid*.

and solely sample from $\theta_1$. This directly compares test-time compute scaling (scaling proof search attempts) to expert iteration scaling (interleaved training on new data sampled from *mathlib-train*) and provides a very clear visualization of the gains of expert iteration. For a fair comparison, we also report an *adjusted compute* line which approximates the test-time performance we would get at each iteration if we were to focus all the additional compute used by expert iteration (sampling proofs from *mathlib-train* as well as re-training models at each iteration) towards solely running proof searches against *mathlib-valid*.

As shown by Figure 2, the scaling exponent of expert iteration is substantially higher than the scaling exponent associated with solely scaling test-time compute (running more proof searches), demonstrating the clear benefit of expert iteration. We'll denote the fully iterated model from this section as $\theta_9^{mathlib}$.

Even in the presence of ground-truth proofs for each of the statements in *mathlib-train* (`tactic` dataset), expert iteration generates data that further improves the performance of the model. The number of statements proved in *mathlib-train* goes from $17390$ ($67.8\%$) at iteration 1 to $19476$ ($76.0\%$) at iteration 9, while the average proof length of these statements goes from $4.8$ to $4.0$. We hypothesize that this continuously improving performance through expert iteration stems from two effects: (i) the model finding new original proofs for the same statements and (ii) the model closing marginally harder statements at each iteration – which in turn provides more useful training data for the next iteration. By iteration 9, the model is trained on more than $90\%$ generated data. We present in Appendix I a few examples of original proofs found by our models on *mathlib-train* compared with their ground-truth versions.

To verify our hypothesis that expert iteration is capable of closing a curriculum of increasingly difficult problems out of a set of problem statements, and that this capability is independent of having access to ground-truth proofs, we propose in the next section to study expert iteration applied to a synthetically generated set of problems for which we have fine-grained control on the difficulty of each statement.

## 5 STATEMENT CURRICULUM LEARNING

In this section we focus on running expert iteration on synthetic statements generated by an inequality generator. The use of synthetic statements enables us to control the difficulty of each statement to present evidence that expert iteration can hill-climb the intrinsic difficulty gradient of the resulting set

of statements. In particular, we show that, at fixed compute budget, expert iteration eventually closes proofs of hard statements that remain completely out of reach of simply sampling proof searches without interleaved training.

## 5.1 Synthetic inequality generator

We designed a synthetic inequality statement generator for Lean in the spirit of the INT (Wu et al., 2021) generator. The generator consists in generating inequalities from well known inequality theorems (AM-GM, Trivial inequality, Cauchy-Schwarz, Bernoulli, Young, Hölder) and composing them. It is driven by two difficulty parameters: $N_D$ which controls depth of composition of inequalities and $N_S$ which controls the complexity of the input expressions to the composed inequalities. We provide details on its implementation in Appendix F.

Using this generator we generate a curriculum of 5600 inequality statements (for which we don't have proofs), 100 for each values of $0 \leq N_S \leq 7$ and $0 \leq N_D \leq 6$. We denote this set of statements as *synth-ineq*. To bootstrap our models capabilities on this specific task, we also generate 100 statements of low difficulty ($N_D = 1$ and $N_S = 5$) and formalize a proof for each of these statements. We refer to this dataset as *synth-ineq-train*. In the rest of this paper we adjunct this training dataset to the `tactic` dataset used to train our models.

## 5.2 Expert iteration on synthetic inequality statements

In this section we propose to set $St$ to the union of the statements in *mathlib-train* and *synth-ineq*. Again, we uniformly set $a = 1$ and use $\theta_0$ and $\theta_1$ as described in Section 4.3, except that they are now also trained on *synth-ineq-train*.

Similarly to the previous section, we report in Figure 3 the cumulative pass rate for two loops, our standard expert iteration loop, and a proof search only loop where we do not interleave training between iterations. The pass rates are reported split by values of $N_D$ (pooling together $0 \leq N_S \leq 7$) which we found to be the main driver for difficulty.

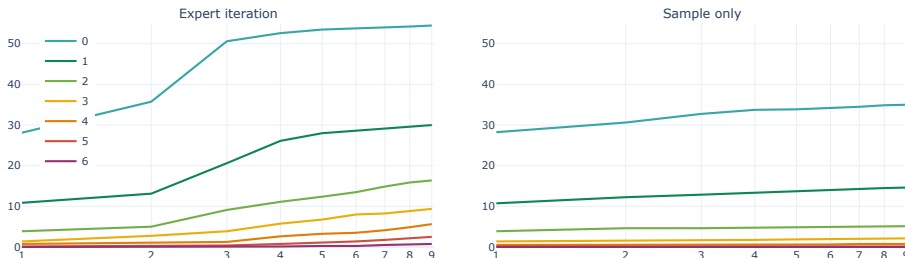

Figure 3: Cumulative pass rate for our *expert iteration* loop as well as a *sample only* loop where we skip re-training the model between iterations. Pass rates are reported for each value of $N_D$ (pooling together $0 \leq N_S \leq 7$).

Despite the challenging nature of these synthetic inequalities, Figure 3 demonstrates that expert iteration is capable of learning the intrinsic curriculum induced by *synth-ineq*. In particular, expert iteration is capable of closing 6 problems of difficulty $N_D = 6$ without having been provided with any seed ground-truth proof for this difficulty level. Note that difficulty $N_D = 6$ remains completely out of reach of simply scaling the number of attempts per statements (the *sample only* loop remaining stuck at 0 for $N_D = 6$).

This confirms on our synthetic statements dataset *synth-ineq* that not only expert iteration is capable of learning the curricula occurring in a set of statements, but this process also enables the emergence of new capabilities without the need for ground-truth proofs (ability to close, highly challenging, deeply composed inequalities).

# 6 TARGETING *miniF2F*

Motivated by the results from Section 5, we curated and manually formalized a set of math exercises denoted as *miniF2F-curriculum* to target *miniF2F*. *miniF2F-curriculum* contains 327 statements from various sources, with their provenance and analysis detailed in Appendix G.

*miniF2F* statements being quite out of distribution compared to *mathlib* statements (which typically are generic theorems and lemmas), we hypothesized that if the difficulty of *miniF2F-curriculum* was made varied enough, expert iteration could potentially leverage it to effectively shift our models' distribution closer to *miniF2F*'s, and in turn, improve their eventual performance on it.

## 6.1 TRANSFER TO *miniF2F*

In this section we propose to set $St$ to the union of the statements in *mathlib-train*, *synth-ineq* and *miniF2F-curriculum*. We uniformly set $a = 1$ on *mathlib-train* and *synth-ineq* and $a = 8$ on *miniF2F-curriculum* and use $\theta_0$ and $\theta_1$ as described in Section 5.

Similarly to previous sections, we report in Figure 4 (left) the cumulative pass rate on *miniF2F-valid* of our full curriculum expert iteration loop and compare them with the *mathlib-train* only expert iteration from Section 4.5. Since more compute is deployed in our full-curriculum loop (more statements), we also report a *mathlib-train* only loop taking $a = 2$. At the end of the expert iteration, 100 out of the 327 statements from *miniF2F-curriculum* end up being closed, suggesting a lack of density in our manually formalized set of statement.

We also report in Figure 4 (right) the *pass@1* and *pass@8* for our full curriculum expert iteration loop. The steady improvement on *miniF2F-valid* shows that the expert iteration procedure we propose does not overfit on the statements that compose the curriculum it uses. Despite the potential inefficiency of our curriculum, the improved performance associated with its use demonstrates, as hypothesized, an effective transfer between *miniF2F-curriculum, synth-ineq* and *miniF2F-valid* through expert iteration. We will denote the fully iterated model from this section as $\theta_9^{full}$.

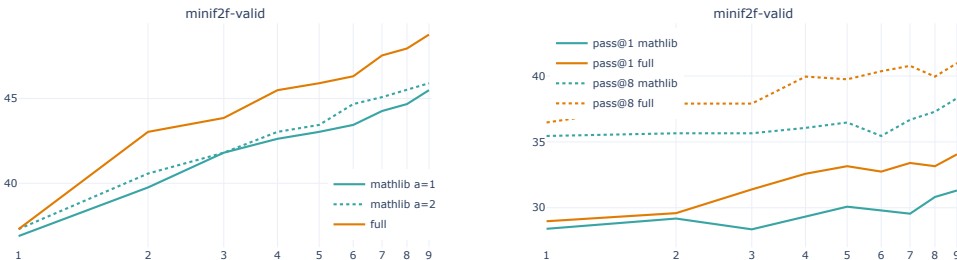

Figure 4: **Left:** cumulative pass rate on *miniF2F-valid* for our expert iteration loop using our full curriculum (*mathlib-train*, *synth-ineq* and *miniF2F-curriculum*) compared to the expert iteration loop from Section 4.5. The total number of attempts per iteration in our *full* loop is $25k + 5.6k + 8 * 327 \approx 33.2k$, which means the total compute deployed is higher than in the *mathlib-train* only loop ($25k$). We therefore also report in dotted a *mathlib-train* only loop, taking $a = 2$, whose total number of attempts per iteration is $\approx 50k$. **Right:** *pass@1* (plain) and *pass@8* (dotted) for our expert iteration loop using our full curriculum (*mathlib-train*, *synth-ineq* and *miniF2F-curriculum*) compared to the *expert iteration* loop from Section 4.5.

## 6.2 RESULTS

We report in Table 2 the pass rates on *mathlib-{valid, test}* and *miniF2F-{valid, test}* for the models trained in previous sections, namely $\theta_1$, $\theta_9^{mathlib}$, and $\theta_9^{full}$. We achieve a $47.3\%$ pass rate (using $a = 64$ attempts) on *miniF2F-valid* and a $36.6\%$ pass rate on *miniF2F-test*, substantially improving from the previous state-of-the-art (Zheng et al., 2022).

These results include the resolution of 26 AMC12 problems, 6 AIME problems and 2 IMO-adapted problems. Out of these statements, 4 AMC12 problems (amc12b_2020_p5, amc12a_2009_p9,

Table 2: Performance of $\theta_1$ (value-function based search), $\theta_9^{mathlib}$ (expert iterated on *mathlib-train*) and $\theta_9^{full}$ (expert iterated on our full curriculum) on *mathlib-{valid, test}* and *miniF2F-{valid, test}*. All proof searches are run with $d = 512$ and $e = 8$.

| Model | pass@1 | pass@8 | pass@64 | pass@1 | pass@8 | pass@64 |
|---|---|---|---|---|---|---|
| *mathlib-valid* | | | | *mathlib-test* | | |
| PACT (Han et al., 2022) | 48.4% | - | - | - | - | - |
| $\theta_1$ | 56.3% | 66.3% | 72.0% | 56.5% | 66.9% | 73.7% |
| $\theta_9^{mathlib}$ | **62.6%** | **70.7%** | **75.8%** | **63.0%** | 71.5% | **77.1%** |
| $\theta_9^{full}$ | 61.7% | 69.8% | 75.3% | 62.9% | **71.6%** | 76.3% |
| *miniF2F-valid* | | | | *miniF2F-test* | | |
| PACT (Zheng et al., 2022) | 23.9% | 29.3% | - | 24.6% | 29.2% | - |
| $\theta_1$ | 28.5% | 35.5% | 41.2% | 25.9% | 31.1% | 33.6% |
| $\theta_9^{mathlib}$ | 31.3% | 38.3% | 44.1% | 27.2% | 33.0% | 35.2% |
| $\theta_9^{full}$ | **33.6%** | **41.2%** | **47.3%** | **29.6%** | **34.5%** | **36.6%** |

amc12a_2003_p24, amc12b_2003_p17), 2 AIME problems (aime_1984_p1, aime_1990_p4), and 2 IMO-adapted problems (imo_1961_p1[2], imo_1964_p2) are uniquely solved by expert iterated models, the two IMO-adapted and the two AIME problems being uniquely solved by $\theta_9^{full}$.

We provide a selection of the proofs found by our models for these statements as well as a qualitative analysis of them in Appendix J. Also, we achieve a new state-of-the-art: higher than $75\%$ pass rate (using $a = 64$ attempts) on *mathlib-{valid, test}*, suggesting that our models could potentially be effectively leveraged as proof assistants in the formalization efforts associated with *mathlib*.

# 7 DISCUSSION AND LIMITATION

Throughout this paper, we used a single model size (774m trainable parameters). We refer readers to Appendix H for more discussion on model size, compute budget and training time. Despite our models' capability, as discussed in Appendix J.1, to generate cuts and witnesses, we believe that their current main limitation lies in their inability (under our proposed search procedure) to chain more than 2 or 3 non-trivial steps of mathematical reasoning, preventing them from consistently solving challenging olympiad problems. We've been repeatedly impressed by the complexity of some of the proofsteps generated by our models. But, proofs requiring many of such reasoning steps remain beyond our current compute horizon. Even if we solved a selection of challenging olympiad problems, our models are still far from being competitive with the brightest students in these competitions.

While our models have demonstrated some capabilities to generate cuts, the cuts they generate are often shallow (they involve only a few proofsteps and don't necessarily deeply change the structure of the proof–we refer the reader to the Cut-Elimination theorem and Carbone & Semmes (1996) for a discussion of the influence of cuts on proof size). We believe that studying language models' ability to generate cuts, and designing search procedures that leverage that capability (related ideas can be found in Czechowski et al. (2021)), are interesting avenues of research to alleviate this limitation.

# 8 CONCLUSION

In this paper we presented an expert iteration procedure for *GPT-f* (Polu & Sutskever, 2020), demonstrating that it is capable of solving a curriculum of increasingly difficult problems out of a set of formal statements of sufficiently varied difficulty. Our results suggest that the lack of self-play in the formal mathematics setup can be effectively compensated for by automatically/manually curated sets of formal statements, which are much cheaper to formalize than full proofs. Finally, we hope that the *statement curriculum learning* methodology we presented in this work will help accelerate progress in automated reasoning, especially if scaled with automated generation and curation of formal statements in the future.

---

[2]This IMO-adapted statement from *miniF2F-valid* is a much weaker version than the original problem (see Appendix J for more context).

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

## A    RELATED WORK

**Deep learning applied to premise selection and proof guidance**    Early applications of deep learning to formal mathematics focused primarily on premise selection and proof guidance. Deep-Math (Irving et al., 2016) explored the use of CNNs and RNNs to predict whether a premise is useful to demonstrate a given conjecture. Their results were later improved with FormulaNet (Wang et al., 2017) by the use of graph neural networks, reminiscent of NeuroSAT (Selsam et al., 2019). Proof guidance consists in selecting the next clause to process *inside* an automated theorem prover. Loos et al. (2017) investigated the use of models similar to DeepMath's for proof guidance and demonstrated a significant uplift on the Mizar library. More recently Firoiu et al. (2021) demonstrated the potential of deep learning techniques to be competitive with E prover's heuristics when applied to resolution calculus while training on fully synthetic data.

**Deep learning applied to automated theorem-proving**    *HOList* (Bansal et al., 2019b) proposes a formal environment based on HOL Light. They achieve their best performance (Bansal et al., 2019a) with a GNN model designed for premise selection and the use of exploration. The same team studied the use of a skip-tree objective with Transformers on formal statements (Rabe et al., 2021), demonstrating, along with GPT-f (Polu & Sutskever, 2020), the potential of leveraging Transformers for formal reasoning. *GamePad* (Huang et al., 2019) and *CoqGymn/ASTactic* (Yang & Deng, 2019) introduce environments based on the Coq theorem prover. *ASTactic* generates tactics as programs by sequentially expanding a partial abstract syntax tree. Urban & Jakubuv (2020) studied the capability of GPT-2 to produce useful conjectures for the Mizar library and IsarStep (Li et al., 2021) explored the synthesis of intermediate propositions in declarative proofs for Isabelle/HOL using Transformers.

**Targeting miniF2F**    Lample et al. (2022) designed HyperTree Proof Search (HTPS), an online training procedure targeting Lean, Metamath and hand-crafted environment named Equations. Lample et al. (2022) report 41% pass-rate on *miniF2F-test* and 42.5% pass-rate on *miniF2F-curriculum* in Lean (de Moura et al., 2015; lea) setup. Thor (Jiang et al., 2022) combined language model and Sledgehammer (Paulson, 2010) and achieved 29.9% pass-rate on *miniF2F-test* in Isabelle setup, which is later improved to 35.2% by Wu et al. (2022) leveraging autoformalization and expert iteration.

## B    LEAN-GYM

`lean-gym` presents the following API:

- init-search: $declaration \rightarrow tactic\_state$. Takes a declaration name (a theorem name from the loaded library) and initializes a search while setting the run-time environment at that particular declaration. It returns the initial tactic state along with a fresh `search_id` and `tactic_state_id`.
- run_tac: $(tactic\_state, tactic) \rightarrow tactic\_state$. Takes a `search_id` and a `tactic_state_id` to identify a tactic state, as well as a tactic string to apply to it. It returns a new tactic state and its associated `tactic_state_id`.

Below is an example in-terminal trace demonstrating the use of `lean-gym`'s REPL interface:

```
$ lean --run src/repl.lean
["init_search", ["int.prime.dvd_mul", ""]]
{
  "error":null,
  "search_id":"0",
  "tactic_state":"⊢ ∀ {m n : ℤ} {p : ℕ}, nat.prime p →
                    ↑p | m * n → p | m.nat_abs ∨ p | n.nat_abs",
  "tactic_state_id":"0"
}
...
["run_tac",["1","1","apply (nat.prime.dvd_mul hp).mp"]]
{
```

```
"error":null,
"search_id":"1",
"tactic_state":"m n : ℤ, p : ℕ, hp : nat.prime p, h : ↑p | m * n
                ⊢ p | m.nat_abs * n.nat_abs",
"tactic_state_id":"2"
}
...
```

Using `lean-gym` is virtually equivalent to opening a Lean editor at a specific theorem, deleting its proof and interacting with Lean to reconstruct it.

Providing a REPL interface over the standard input/output makes it very easy to integrate `lean-gym` from any programming language. Writing a wrapper in Python, as an example, only takes a few dozen lines of code. Since `lean-gym` is a Lean program, managing the loaded libraries is done directly using Lean's own infrastructure (using `leanpkg.toml`), making it quite straightforward to have access to both *mathlib* and *miniF2F* statements from the same `lean-gym` instance.

Note that `lean-gym` is stateful, meaning that distributing proof searches on multiple `lean-gym` instances requires to track which instance is associated with which proof search. In practice, we were able to scale the use of `lean-gym` to thousands of cores running thousands of proof searches in parallel. Finally, `lean-gym`'s REPL interface is blocking, preventing inner-proof search parallelization, though this limitation can probably be removed in the future.

## C  WEBMATH

Our updated *WebMath* pre-training dataset consists in the mix presented in table 3.

Table 3: Mix and source of data involved in the updated *WebMath* pre-training.

| Dataset | Size | Mix |
|---|---|---|
| Github Python | 179 GB | 25% |
| arXiv Math | 10 GB | 25% |
| Math StackExchange | 2 GB | 25% |
| PACT `mix2` | 28 GB | 17% |
| Math Overflow | 200 M | 5% |
| ProofWiki | 30 M | 2% |
| PlanetMath | 25 M | 1% |

As demonstrated in table 3, we empirically up-weighted (compared to their token size) parts of *WebMath* with high-quality mathematical content while making sure they don't overfit (despite running >1 epochs for some of them). We also included PACT `mix2` directly in the *WebMath* pre-training to avoid having to sequence more than two pre-training phases to prepare Lean models.

## D    EXAMPLE OF MINIF2F INPUT, LEAN ENVIRONMENT AND MODEL OUTPUT

We illustrate an example of the interaction between Lean environment and our model. In the figure shown below, the model has 1 output for each current goal (corresponding to 1 expand budget). The model could have expand budget bigger than 1, in which case the search procedure becomes a tree.

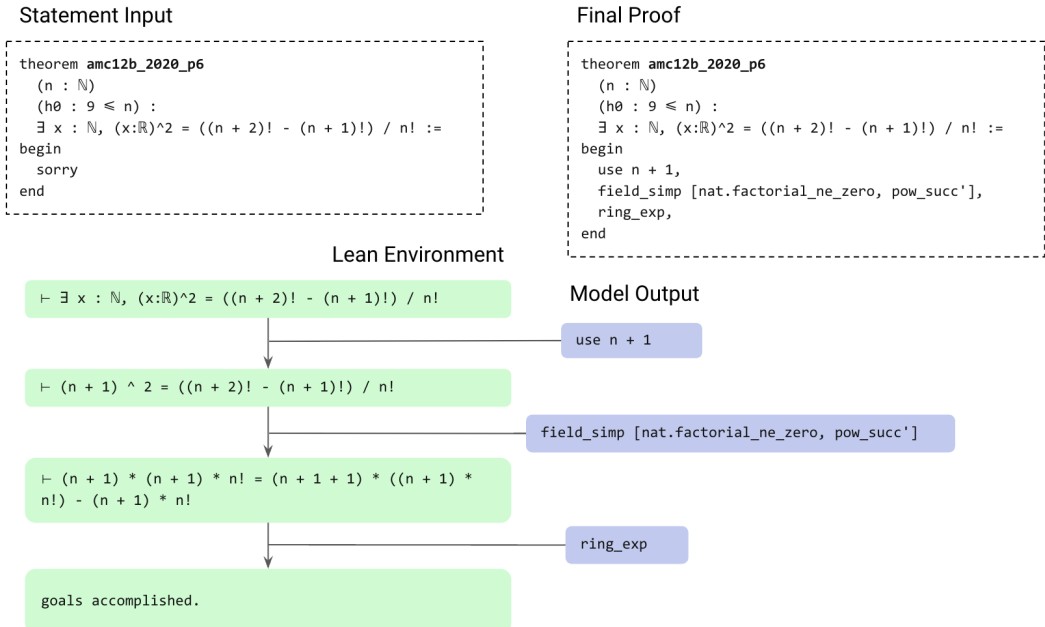

Figure 5: Input from *miniF2F* consists of a mathematical statement written in formal language (here the Lean version) without proof. Lean environment parses the statement and exposes to users the goal to be proved. The model outputs a line of code (tactics and corresponding arguments). Lean environment receives the model output and transforms the previous goal to another goal to be proved. This process is repeated till all remaining goals are closed. In this case, the original statement is proved: the final proof is collected by following the trajectory of model's output.

## E    ILLUSTRATION OF EXPERT ITERATION

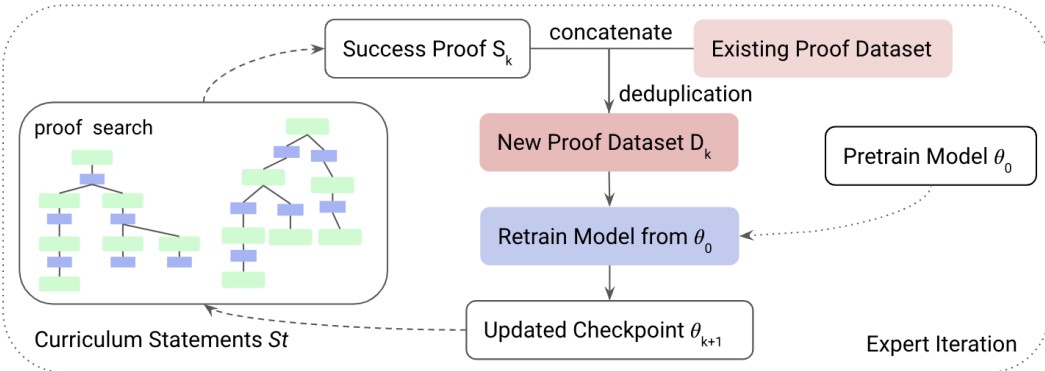

Figure 6: Illustration of expert iteration. The notation in this figure corresponds to Section 4.4 in main text.

# F   SYNTHETIC INEQUALITIES

## F.1   DESIGN

The generator consists of three phases:

**Seed expressions generation**   The first phase consists in generating seed expressions for which we track the sign. We start by initializing an expression set $E$ composed of tuples of expressions and sign constraints, by generating $n_v$ variable names (letters) assumed strictly positive as well as $n_n$ integers (for which we know the sign). For $N_S$ rounds, we compose elements of $E$ using unary $(log(\cdot), log(1/\cdot), sqrt(\cdot))$ or binary operations $(+, -, \times, /, \wedge, max, min)$ for which we can deduce the sign based on the sign condition of the input expression(s) and re-inject the resulting expression and sign constraint in $E$. This produces a set $E$ of signed seed expressions of size $n_v + n_n + N_S$.

**Inequality composition**   The second phase consists in generating inequalities from well known inequality theorems (AM-GM, Trivial inequality, Cauchy-Schwarz, Bernoulli, Young, Hölder) taking as input to these theorems expressions from $E$ based on the sign constraints required for each theorem. We finally compose these inequalities $N_D$ times using compositions theorems detailed in F.2. The resulting inequality is a composed inequality of depth $N_D$ based on $n_v + n_n + N_S$ seed expressions.

**Simplification**   We finally post-process these inequalities so that they are parsable by Lean and run them through Lean's `simp` tactic for a final simplification.

$N_D$ and $N_S$ together control for the difficulty of the resulting inequality. $N_D$ controls depth of composition, while $N_S$ controls for obfuscation as it increases the complexity of the input expressions to the composed inequalities. When sampling inequalities, we $n_n = 4$ and randomly sample $2 \leq n_v \leq 8$ at each generation. We report below examples of generated inequalities for various values of $N_D$ and $N_S$.

## F.2   LIST OF INEQUALITY COMPOSITION THEOREMS

Below is the list of theorem names from *mathlib* that we use to compose inequalities together. One third of the time, we only transform the current composed inequality with one of the following theorems:

- `neg_le_neg`
- `inv_le_inv`
- `mul_self_le_mul_self`
- `div_le_one_of_le`

We otherwise compose the current composed inequality with a newly generated inequality using the following theorems:

- `mul_le_mul`
- `add_le_add`
- `div_le_div`
- `mul_le_mul_of_nonneg`
- `le_mul_of_ratio`

## F.3   EXAMPLES

$N_D = 0 \, N_S = 0$

$N_D = 0 \, N_S = 4$

| Compositions | `AmGm a b (67:ℝ) ((1:ℝ)/(10:ℝ)) ((1:ℝ)/(10:ℝ)) ((8:ℝ)/(10:ℝ))` |
|---|---|
| Statement | `theorem synthetic_ineq_nb_seed_var_0_depth_0_p_1`
`  (a b : ℝ)`
`  (h0 : 0 < a)`
`  (h1 : 0 < b) :`
`  (67:ℝ) ^ ((8:ℝ) / (10:ℝ)) * b ^ (10:ℝ)⁻¹ *`
`    a ^ (10:ℝ)⁻¹ ≤  (8:ℝ) / (10:ℝ) * (67:ℝ) +`
`    (10:ℝ)⁻¹ * a + b * (10:ℝ)⁻¹ := sorry` |

| Compositions | `Sqnonneg a ((a) + ((-68:ℝ)))` |
|---|---|
| Statement | `theorem synthetic_ineq_nb_seed_var_4_depth_0_p_4`
`  (a b : ℝ)`
`  (h0 : 0 < a)`
`  (h1 : 0 < b) :`
`  (2:ℝ) * (a * (a + -(68:ℝ))) ≤`
`    (a + -(68:ℝ)) ^ 2 + a ^ 2 := sorry` |

$N_D = 4 \, N_S = 4$

| Compositions | `AddLeAdd`
`Bernoulli 99 c`
`AddLeAdd`
`SelfDivConst ((a) / (f)) 6`
`LeMulOfRatio`
`SelfDivConst c 70`
`DivLeDiv`
`Cauchy ((a) / (f)) d c (log (((59:ℝ) + f)))`
`Young ((a) / (f)) a ((3:ℝ)/(2:ℝ)) ((3:ℝ)/(1:ℝ))` |
|---|---|
| Statement | `theorem synthetic_ineq_nb_seed_var_4_depth_4_p_13`
`  (a b c d e f : ℝ)`
`  (h0 : 0 < a)`
`  (h1 : 0 < b)`
`  (h2 : 0 < c)`
`  (h3 : 0 < d)`
`  (h4 : 0 < e)`
`  (h5 : 0 < f) :`
`  (1:ℝ) + (99:ℝ) * c + (a / f / (6:ℝ) + a * (a / f) /`
`    ((d ^ 2 + a ^ 2 / f ^ 2) *`
`    (real.log ((59:ℝ) + f) ^ 2 + c ^ 2))) ≤`
`    ((a / f) ^ ((3:ℝ) / (2:ℝ)) / ((3:ℝ) / (2:ℝ)) +`
`    a ^ 3 / (3:ℝ)) /`
`    (real.log ((59:ℝ) + f) * d + a / f * c) ^ 2 *`
`    (c / (c / (70:ℝ))) + a / f + (c + (1:ℝ)) ^ 99`
`  := sorry` |

## G    MINIF2F-CURRICULUM

The 327 statements of *miniF2F-curriculum*[3] are manually formalized from:

- **AOPS Books (Lehoczky & Rusczyk, a;b)**: 302 examples and exercises. The books are classic problem solving textbooks for students in grades 7-12 preparing for contests such as AMCs and AIMEs. We skipped problems that were too challenging to formalize due to missing infrastructure in *mathlib* or non-suitable format for formalization (see section *Formalization effort and challenges* in Zheng et al. (2022)).
- **MATH (Hendrycks et al., 2021) dataset**: 25 problems. All problems were drawn from the train split of the dataset, focusing on difficulty 5 problems (*miniF2F* only contains problems from the test split).

We verified (based on problem provenance and manual inspection of statements) that *miniF2F-curriculum* had an empty intersection with *miniF2F-{test, valid}*. We refer to Zheng et al. (2022) for more details on the formalization procedure and the typical time needed for it as these problems were formalized in similar conditions.

## H    MODEL SIZE

Other than the single model size we use in the experiment reported in the main text (774m trainable parameters), we briefly experimented with different model sizes (not reported in this paper) and found that model size scaling is not as straightforward as in the case of unsupervised learning (Kaplan et al., 2020). We found that bigger models are better, in the sense that they consistently exhibit higher *pass@1*. But, they are also much more expensive to sample from. And despite their *pass@1* being higher, it is often the case that for a fixed amount of compute, sampling more attempts from a smaller model leads to a better final performance.

For the compute budget we had available, we estimated the model size we used to be a compelling trade-off. We leave as future work a more thorough study of these dynamics to better understand the different compute frontiers involved. Indicatively, with our 774m parameters model, running a full expert iteration to train $\theta_9^{full}$ required about 2000 A100 days of compute. Running one full proof search ($a = 1\ d = 512\ e = 8$) when properly parallelised, requires on average about 0.1 A100 hour of compute.

---

[3]`https://github.com/openai/miniF2F/tree/statement_curriculum_learning/lean/src/`
`statement_curriculum_learning`

## I    EXAMPLE PROOFS FROM *mathlib-train*

We present in this section original proofs found by our models from *mathlib-train*, compared with their ground-truth version.

`comap_eq_of_inverse`

| | |
|---|---|
| Statement | ```lemma comap_eq_of_inverse {f : filter α} {g : filter β}`
`{φ : α → β} (ψ : β → α) (eq : ψ ∘ φ = id)`
`(hφ : tendsto φ f g) (hψ : tendsto ψ g f) :`
`comap φ g = f :=``` |
| Ground-truth | ```begin`
`  refine ((comap_mono $`
`    map_le_iff_le_comap.1 hψ).trans _).antisymm`
`    (map_le_iff_le_comap.1 hφ),`
`  rw [comap_comap, eq, comap_id],`
`  exact le_rfl`
`end``` |
| Model proof | ```begin`
`  refine le_antisymm _ (filter.map_le_iff_le_comap.1 hφ),`
`  refine λ s hs, _,`
`  rw mem_comap,`
`  use [ψ ⁻¹' s, hψ hs],`
`  rw [← preimage_comp, eq, preimage_id]`
`end``` |

sum_range_sub_sum_range

| | |
|---|---|
| Statement | ```lean
lemma sum_range_sub_sum_range {α : Type*}
  [add_comm_group α] {f : ℕ → α} {n m : ℕ}
  (hnm : n ≤ m) :
  ∑ k in range m, f k - ∑ k in range n, f k =
  ∑ k in (range m).filter (λ k, n ≤ k), f k :=
``` |
| Ground-truth | ```lean
begin
  rw [← sum_sdiff (@filter_subset _ (λ k, n ≤ k) _
    (range m)), sub_eq_iff_eq_add,
    ← eq_sub_iff_add_eq, add_sub_cancel'],
  refine finset.sum_congr
    (finset.ext $ λ a, ⟨λ h, by simp at *; finish,
    λ h, have ham : a < m :=
      lt_of_lt_of_le (mem_range.1 h) hnm,
      by simp * at *⟩)
    (λ _ _, rfl)
end
``` |
| Model proof | ```lean
begin
  rw [← sum_Ico_eq_sub _ hnm],
  congr,
  apply finset.ext,
  simp [Ico.mem, *],
  tauto
end
``` |

`prod_inv_distrib`

| | |
|---|---|
| Statement | ```lemma prod_inv_distrib : (∏ x in s, (f x)⁻¹) = (∏ x in s, f x)⁻¹ :=``` |
| Ground-truth | ```begin classical, by_cases h : ∃ x ∈ s, f x = 0, { simpa [prod_eq_zero_iff.mpr h, prod_eq_zero_iff] using h }, { push_neg at h, have h' := prod_ne_zero_iff.mpr h, have hf : ∀ x ∈ s, (f x)⁻¹ * f x = 1 := λ x hx, inv_mul_cancel (h x hx), apply mul_right_cancel' h', simp [h, h', ← finset.prod_mul_distrib, prod_congr rfl hf] } end``` |
| Model proof | ```begin classical; induction s using finset.induction_on with a s has ih, { simp, }, simp only [has, prod_insert has, mul_inv_rev'], finish end``` |

## J    EXAMPLE PROOFS FROM *miniF2F-{test, valid, curriculum}*

We present in this section proofs found by our models from *miniF2F-{test, valid, curriculum}*, demonstrating some of the capabilities emerging from our training procedure.

### J.1    QUALITATIVE ANALYSIS OF PROOFS

We provide qualitative insights in the nature of the proofs found by our models, which we believe are useful to build a better intuition of their capabilities beyond pass rate numbers. Throughout this section, we refer to statements and solutions found by our models that are presented in Appendix J along with comments describing the specificity of each proof.

First, we observe that a large number of olympiad problems that are designed to be computationally challenging for humans are rendered trivial for our models through the use of Lean tactics. As an example, `mathd_numbertheory_447` which is not necessarily considered straightforward for humans, can be closed in Lean by a simple `refl` (proof found by our models).

In recent years, Lean's `mathlib` community has developed high-powered tactics such as `linarith/nlinarith` (solves (non)linear inequalities), `norm_num` (normalizes numerical expressions), `simp` (simplifies goals and hypotheses) and `ring` (normalizes expressions in a ring). These tactics can be used with arguments to guide their underlying search procedure. As mentioned in Zheng et al. (2022), we confirm here that our models acquire advanced capabilities to leverage these high-level tactics by providing exogenous arguments which are not present in the current tactic state. The generation of these exogenous arguments through language modeling seems to require a non-trivial amount of mathematical intuition. `imo_1964_p2`, `imo_1961_p1` and `aime_1990_p15` are good examples of such uses.

We have also observed a number of proofs that require multiple non-trivial reasoning steps through the use of lower-level tactics such as `use`, `have`, or `by_cases` that generally involve producing a witness or chaining implications, requiring the generation of context specific exogenous terms. These interesting reasoning steps are structurally different from simple normalization, simplification and rewriting of hypotheses or goals because they heavily rely on our models ability to generate meaningful cuts or witnesses. This capability is, in our opinion, the most exciting stepping stone towards solving more challenging mathematical problems. See, `aopsbook_v2_c8_ex1`, `amc12b_2020_p6` and `mathd_train_algebra_217` for examples of such proofs.

More generally, we also observe that proofs generated by our models have a distinctive style compared to proofs formalized by humans. This stems in part from the model's capability to leverage high-level tactics in a way that is challenging for humans as discussed in this section (e.g. one-liners such as `nlinarith [sq_nonneg (x - y), sq_nonneg (y - z)]` where humans would generally decompose the problem in a less machine-like way). Additionally, as a result of our search procedure and despite the bias towards shorter proofs introduced by our value function, extraneous proofsteps (such as reversion/introduction of hypotheses, or no-op rewrites) are often interleaved with useful ones, which rarely happens in human formalizations.

`imo_1961_p1`

| | |
|---|---|
| Natural language | Solve the system of equations:

$$x + y + z = a$$
$$x^2 + y^2 + z^2 = b^2$$
$$xy = z^2$$

where $a$ and $b$ are constants. Give the conditions that $a$ and $b$ must satisfy so that $x, y, z$ (the solutions of the system) are distinct positive numbers. **Note**: the formalized statement in *miniF2F* is a weaker problem as it focuses on the second part of the question, providing the actual conditions, and asking for a proof that the requirement entails them. |
| Model proof | ```
theorem imo_1961_p1
  (x y z a b : ℝ)
  (h₀ : 0 < x ∧ 0 < y ∧ 0 < z)
  (h₁ : x ≠ y)
  (h₂ : y ≠ z)
  (h₃ : z ≠ x)
  (h₄ : x + y + z = a)
  (h₅ : x^2 + y^2 + z^2 = b^2)
  (h₆ : x * y = z^2) :
  0 < a ∧ b^2 < a^2 ∧ a^2 < 3 * b^2 :=
begin
  revert_all,
  intros,
  rw mul_comm,
  split,
  { nlinarith [sq_nonneg (x - y), sq_nonneg (y - z)], },
  split,
  { nlinarith [sq_nonneg (z - 1)], },
  revert h₃ h₄,
  field_simp [mul_comm a b],
  rw [mul_comm, ← h₅],
  contrapose!,
  rw mul_comm at h₆,
  rw mul_comm,
  intro h,
  nlinarith [sq_nonneg (x - y), sq_nonneg (y - z)]
end
``` |
| Comments | The model is able to close this problem by splitting into cases, contraposing for the last case and using `nlinarith`. It must be noted that the arguments for the first two `nlinarith` uses are not necessary, however the `[sq_nonneg (x - y), sq_nonneg (y - z)]` argument provided on the last line is crucial to close the goal and are completely exogenous (present in no form in the tactic state before). |

imo_1964_p2

| Natural language | Suppose $a, b, c$ are the sides of a triangle. Prove that $$a^2(b + c - a) + b^2(c + a - b) + c^2(a + b - c) \leq 3abc$$ |
|---|---|
| Model proof | ```
theorem imo_1964_p2
  (a b c : ℝ)
  (h₀ : 0 < a ∧ 0 < b ∧ 0 < c)
  (h₁ : c < a + b)
  (h₂ : b < a + c)
  (h₃ : a < b + c) :
  a^2 * (b + c - a) + b^2 * (c + a - b) + c^2 *
    (a + b - c) ≤ 3 * a * b * c :=
begin
  nlinarith [sq_nonneg (b - a), sq_nonneg (c - b),
    sq_nonneg (a - c), sq_nonneg (c - a)]
end
``` |
| Comments | The model is able to close an IMO problem in one-line. It correctly provides exogenous arguments to nlinarith, which are necessary to close the goal. Note that either one of the last two arguments in the sequence [sq_nonneg (b - a), sq_nonneg (c - b), sq_nonneg (a - c), sq_nonneg (c - a)] can be omitted. |

`aime_1990_p15`

| | |
|---|---|
| Natural language | Find $ax^5 + by^5$ if the real numbers $a, b, x,$ and $y$ satisfy the equations $$ax + by = 3,$$ $$ax^2 + by^2 = 7,$$ $$ax^3 + by^3 = 16,$$ $$ax^4 + by^4 = 42.$$ **Note**: the formalized statement in *miniF2F* provides the answer and asks for a proof of it. |
| Model proof | <pre>theorem aime_1990_p15
  (a b x y : ℝ)
  (h₀ : a * x + b * y = 3)
  (h₁ : a * x^2 + b * y^2 = 7)
  (h₂ : a * x^3 + b * y^3 = 16)
  (h₃ : a * x^4 + b * y^4 = 42) :
  a * x^5 + b * y^5 = 20 :=
begin
  revert_all,
  intros a b x y h₃ h₁ h₂ h₄,
  ring_nf at h₁ h₂,
  rw ← sub_eq_zero at h₁,
  nlinarith [sq_nonneg (x - y), sq_nonneg (a + b - 2),
    sq_nonneg (x + y - (2:ℝ)), sq_nonneg (a - b - 2)]
end</pre> |
| Comments | The model is able to close a challenging AIME problem by providing crucial exogenous arguments `sq_nonneg (x - y)` and `sq_nonneg (x + y - (2 : ℝ))` to `nlinarith`, which are required to close the goal (while the other two can be removed). |

`mathd_train_algebra_217`

| | |
|---|---|
| Natural language | Let $f(x) = Ax + B$ and $g(x) = Bx + A$, where $A \neq B$. If $f(g(x)) - g(f(x)) = B - A$, what is $A + B$? **Note**: the formalized statement in our *curriculum* provides the answer and asks for a proof of it. |
| Model proof | ```theorem mathd_train_algebra_217
  (a b : ℝ)
  (f g : ℝ → ℝ)
  (h₀ : ∀ x, f x = a * x + b)
  (h₁ : ∀ x, f x = b * x + a)
  (h₂ : a ≠ b)
  (h₃ : ∀ x, f (g x) - g (f x) = b - a) :
  a + b = 0 :=
begin
  revert_all,
  intros a b,
  intros f g,
  contrapose!,
  rintro ⟨h₀, ⟨h₁, h₂⟩⟩,
  use (0 : ℝ),
  simp only [sub_eq_iff_eq_add, h₀, mul_zero] at *,
  simp only [*, zero_add],
  norm_num at h₀
end``` |
| Comments | The model is able to close the goal by contraposing, supplying a witness by the use of use  $(0 : \mathbb{R})$ and finally leveraging the simp and norm_num. This example demonstrates the model's ability to chain multiple non-trivial steps of reasoning including the generation of witnesses. |

amc12b_2020_p6

| | |
|---|---|
| Natural language | For all integers $n \geq 9$, the value of $$\frac{(n+2)! - (n+1)!}{n!}$$ is always which of the following? 
 **(A)** a multiple of $4$     **(B)** a multiple of $10$     **(C)** a prime number 
 **(D)** a perfect square     **(E)** a perfect cube. |
| Model proof | ``` theorem amc12b_2020_p6 (n : ℕ) (h₀ : 9 ≤ n) : ∃ x : ℕ, (x:ℝ)^2 = (nat.factorial (n + 2) - nat.factorial (n + 1)) / nat.factorial n := begin revert_all, intros n hn, use n + 1, field_simp [nat.factorial_ne_zero, pow_succ'], ring_exp end ``` |
| Comments | The model exhibits here its ability to implicitly interpret factorial expressions by guessing the correct answer with the use of use `n + 1` and straightforwardly proving its claim right after. |

`mathd_algebra_140`

| Natural language | The expression $24x^2 - 19x - 35$ can be written as $(Ax - 5)(2Bx + C)$, where $A$, $B$, and $C$ are positive numbers. Find $AB - 3C$. **Note**: the formalized statement in *miniF2F* provides the answer and asks for a proof of it. |
|---|---|
| Model proof | ```
theorem mathd_algebra_140
  (a b c : ℝ)
  (h₀ : 0 < a ∧ 0 < b ∧ 0 < c)
  (h₁ : ∀ x, 24 * x^2 - 19 * x - 35 =
    (((a * x) - 5) * ((2 * (b * x)) + c))) :
  a * b - 3 * c = -9 :=
begin
  revert_all,
  rintro a b c h₁,
  rw mul_comm,
  rw ←sub_eq_zero,
  field_simp,
  rw sub_eq_add_neg,
  rw ← eq_neg_iff_add_eq_zero,
  rw [mul_comm, ← sub_eq_add_neg, sub_eq_add_neg,
  ← sub_eq_add_neg],
  rw ←sub_eq_zero,
  simp only [mul_add, neg_mul_eq_neg_mul_symm, mul_comm,
  add_left_comm],
  norm_num,
  simp only [pow_two],
  intro h₀,
  rw ← sub_eq_zero,
  linarith [h₁.2, h₁.1, h₀ 3, h₀ 1, h₀ 2]
end
``` |
| Comments | The model mostly "struggles" to make progress up to the last line. There, it presents us with the idea to specialize the statement given in $h_1$ (which became $h_0$ in the process of the proof) at three consecutive natural numbers $1, 2, 3$ which closes the goal with `nlinarith`. This proof is interesting as it demonstrates the model's ability to evaluate symbolic expressions implicitly. |

`aime_1984_p1`

| | |
|---|---|
| Natural language | Find the value of $a_2 + a_4 + a_6 + a_8 + \ldots + a_{98}$ if $a_1, a_2, a_3 \ldots$ is an arithmetic progression with common difference 1, and $a_1 + a_2 + a_3 + \ldots + a_{98} = 137$. **Note**: the formalized statement in *miniF2F* provides the answer and asks for a proof of it. |
| Model proof | ```
theorem aime_1984_p1
  (u : ℕ → ℚ)
  (h₀ : ∀ n, u (n + 1) = u n + 1)
  (h₁ : ∑ k in finset.range 98, u k.succ = 137) :
  ∑ k in finset.range 49, u (2 * k.succ) = 93 :=
begin
  revert_all,
  simp [finset.sum_range_succ],
  ring,
  simp [pow_succ, mul_comm, add_left_comm, add_assoc],
  simp [two_mul],
  rintro p,
  revert p,
  simp [add_comm],
  intros p hp,
  simp [hp], ring_nf,
  intros,
  nlinarith
end
``` |
| Comments | The model finds a rather short proof of the statement compared to typical human formalizations (see `https://gist.github.com/MantasBaksys/3efd5c0d42f440d16c09076db42822f2`). |

`aopsbook_v2_c8_ex1`

| | |
|---|---|
| Natural language | Consider the sequence $\{\frac{n}{n+1}\} = \frac{1}{2}, \frac{2}{3}, \frac{3}{4}, \dots$ 
 Prove that this sequence tends to 1 as $n \to \infty$ |
| Model proof | ```
theorem aopsbook_v2_c8_ex1
  (u : ℕ → ℝ)
  (h₀ : ∀ n, u n = n / (n + 1)) :
  filter.tendsto u filter.at_top (𝒩 1) :=
begin
  revert_all,
  simp [← nnreal.coe_one],
  norm_cast,
  intros,
  revert h₀,
  assume h,
  simp [tendsto_const_nhds, ← nnreal.coe_one, h],
  revert u,
  assume f,
  norm_num,
  rw tendsto_iff_norm_tendsto_zero,
  assume H,
  convert tendsto_norm_zero.comp
    tendsto_one_div_add_at_top_nhds_0_nat,
  funext n,
  have h₀ : (((n:ℝ) + 1) : ℝ) ≠ 0,
  { norm_cast, exact n.succ_ne_zero, },
  rw H,
  field_simp [h₀, norm_neg],
  ring,
  rw [← sub_eq_zero],
  simp [← sub_eq_zero],
  simp [sub_eq_add_neg],
end
``` |
| Comments | An interesting example of a generated have statement, denoted by $h_0$, which is introduced as a cut in order to simplify the expression containing divisions by using `field_simp[h₀, norm_neg]` at a later step. |

`mathd_numbertheory_447`

| | |
|---|---|
| Natural language | What is the sum of the units digits of all the multiples of 3 between 0 and 50? **Note**: the formalized statement in *miniF2F* provides the answer and asks for a proof of it. |
| Model proof | ```
theorem mathd_numbertheory_447 :
  ∑ k in finset.filter (λ x, 3|x)
  (finset.erase (finset.range 50) 0), (k % 10) = 78 :=
begin
  refl
end
``` |
| Comments | Because the predicate $\lambda$ x, 3\|x is registered as decidable over $\mathbb{N}$, we can state the problem by using `finset.filter`, which is computable. Hence, `refl` is able to close the goal. |

