# OpenReview forum: "Formal Mathematics Statement Curriculum Learning"
_ICLR.cc/2023/Conference — ICLR 2023 notable top 25%_

### Official Review · Reviewer_kECs · 2022-10-24

**Confidence:** 4
**Correctness:** 4
**Technical Novelty And Significance:** 3
**Empirical Novelty And Significance:** 3
**Recommendation:** 8

**Clarity, Quality, Novelty And Reproducibility:**

I think the approach is novel and provides a good contribution to the state of the art of applying deep learning techniques to automated reasoning.

**Strength And Weaknesses:**

- I enjoyed reading this paper. There is a rich body of work around using RL algorithms to generate proofs, and this is a welcome addition to the literature.
- A general challenge in using RL for theorem proving seems to be that the MDP induced by the possible proof tree changes between theorems, and it can be difficult for a learning system to latch on to consistent features the way it would for a game such as chess or go. As such, it has been difficult for these systems to exhibit superhuman performance.

**Summary Of The Paper:**

This paper considers the problem of theorem proving, and how such a problem can be addressed with RL. Treats the development of a proof as a search through a tree of possible proofs. This results in an MDP in which the actions at each state of the proof tree are the possible proof rules that can be applied.

The paper considers that a direct application of RL methods to proof development is too difficult, given the infinite action space of proof-development. An additional difficulty is that there is no direct way to set up a self-play environment, as in 2-player games. The authors address the first challenge by building on existing work, which samples logical formulas from a transformer model.

The authors address the goal of providing the prover with auxiliary sets of problem statements of varying difficulties. The key result is that when this set of auxiliary set of problem statements is sufficiently varied in difficulty, the trained expert iteration procedure is able to eventually generalize to the target distribution of problems.

The authors use expert iteration: proof search interleaved with learning. As applied in two-player games, this means models are trained on their previously sampled trajectories to achieve continuous improvement. The authors modify this technique to the domain of proof search.

This outperforms proof search alone. Expert iteration is found to be capable of solving a curriculum of problem. Applied to surpass previous best performance on the miniF2F benchmark.

The authors implement their system as a program that chooses proofsteps in the Lean prover. They find improvements over (benchmark). They find, however, that the cut-generation capabilities of their current transformer does not result in formulas that dramatically change the structure of the proof, and suggest further investigation on this point.



**Summary Of The Review:**

I think the paper will be a meaningful contribution to the state of the art.

---

> ### Author Response · Authors · 2022-11-11
> **Response**
>
> Thank you for your review and your time dedicated to this.
> > A general challenge in using RL for theorem proving seems to be that the MDP induced by the possible proof tree changes between theorems, and it can be difficult for a learning system to latch on to consistent features the way it would for a game such as chess or go. As such, it has been difficult for these systems to exhibit superhuman performance.
>
> Thank you for sharing your thoughts. We agree with the reviewer that the characteristics of theorem proving: having in principle infinite statements to prove does pose a challenge to application of RL methods.
>
> In formal math, the agent is not playing against an opponent but against different statements, while a proof tree from one statement could exhibit different underlying features from another given the varying specific domain (e.g. algebra, geometry and number theory).
>
> Therefore, it is interesting to see how RL algorithms are specially tailored for formal math. Our work is in that spirit by adapting expert iteration through a curated set of curriculum statements.

---

### Official Review · Reviewer_GFD4 · 2022-10-25

**Confidence:** 4
**Correctness:** 2
**Technical Novelty And Significance:** 2
**Empirical Novelty And Significance:** Not applicable
**Recommendation:** 5

**Clarity, Quality, Novelty And Reproducibility:**

The paper is not clearly written and organized, making the readers hard to follow. The concept of expert iteration is not novel in the related work, and the paper fails to discuss the contributions of the proposed method.

**Details Of Ethics Concerns:**

Hi, thank you for your responses. They addressed some of my concerns. I'd like to raise my score.


**Strength And Weaknesses:**

**Strengths**

The paper addresses a meaningful problem that investigates if a proof search interleaved with learning would benefit the task of formal mathematics.


**Weaknesses**

**1. The paper is not well organized and hard to follow.**

For example, the abstract is too concise to include necessary information regarding the background and motivation of the problem that the paper is going to solve and the introduction of the proposed method. The introduction lacks a clear and coherent line to follow. I am interested in seeing 1) the typical task of theorem proving and proof search; 2) the limitations of existing methods in theorem proving; 3) the introduction of expert iteration with an illustration of the target task; 4) the design of the proposed method in this paper; 5) the designs of the experiments and the main results. I am not very comfortable with the organization of the related work section as most of the content is put in the appendix. In the methodology section, it would be better to give an example of the miniF2F benchmark and an illustration of the Lean environment. I am struggling to understand the task, the dataset, and the environment that the work is working on.

**2. The experiments are not extensive enough.**

Most of the experiments are conducted on the miniF2F dataset, which consists of 244 validation and 244 test formalized statements of mathematical problems. However, miniF2F is limited to a small data scale, making the results not solid enough. Also, the paper fails to compare more baselines or search strategies in the experiments.

**3. The writing could be improved.**

It would be nice to provide a reference when mentioning some work for the first time. For example, the paper misses the reference when mentioning Go on page 2 and misses the reference for Lean in the related work section. There are some typos in the paper. For instance, "Proof datasets extraction". The statement "These two differences make a naive application of reinforcement learning to formal mathematics unlikely to succeed." lacks the necessary supporting facts in the paper.


**Summary Of The Paper:**

This paper investigates the effectiveness of expert iteration for language models in the task of formal mathematics. The paper shows several advantages of expert iteration in theorem proving: (1) proof search interleaved with learning outperforms proof search alone; (2) expert iteration can solve a curriculum of increasingly difficult problems without associated ground-truth proofs; and (3) expert iteration can beat the previous state-of-the-art on the miniF2F benchmark.

**Summary Of The Review:**

The paper could be improved in different aspects, including writing, organizing, and experiments. It would be nice if the authors could address the concerns I have in the comments above.

---

> ### Author Response · Authors · 2022-11-11
> **Response (1/2)**
>
> Thank you for your review and your time dedicated to this. We hope that our answer clears up some misunderstandings, clarifies key differences to prior work and convinces you of the novelty and relevance of your contribution. We now address the concerns of the reviewer.
> ## Task, Motivation, Design.
> Thank you for the suggestions and we’ve updated a revised version. We now clarify specific questions:
> > the typical task of theorem proving and proof search
>
> To be specific, our work focuses on theorem proving in formal math (Lean system in this paper), which is equipped with an interactive proof assistant. Our work pushes forward the *neural theorem proving*, i.e. leverage deep learning techniques for formal math.
>
> The typical task is to generate a machine-checkable proof given a formal statement without proof. Search strategy is often used during the generation process, which is often termed proof search.
>
> We include examples for the models output, from both mathlib and miniF2F (c.f. Appendix I and J in revised version). In these examples, the model is given only the formal statements without proof, while the proof (between `begin` and `end`) is model-generated. We also add an illustration in Appendix D.
>
> This paper also acts as an effort to push forward the IMO Grand Challenge [A1] promoted in the Lean community: Build an AI that solves IMO problems.
>
> > the limitations of existing methods in neural theorem proving in formal math
>
> The limitations of existing methods, especially the family of state-of-the-art methods using large language models [B2, B3], is the requirement of a large corpus of statements with proof. We have added this point into our revised version.
>
> > In the methodology section, it would be better to give an example of the miniF2F benchmark and an illustration of the Lean environment.
>
> Thank you for your suggestion. Due to page limit, we add an illustration of an example of target task, miniF2F benchmark, Lean environment in Appendix D, and an illustration of expert iteration in Appendix E.
>
> > the introduction of expert iteration with an illustration of the target task. the design of the proposed method in this paper. The designs of the experiments and the main results.
>
> Thank you for your suggestion. Due to page limit, we add an illustration of expert iteration in Appendix E. Please refer to section 4 for an overall design of our method in formal math (pretraining, bootstrapping, expert iteration) targeting mathlib-valid. This design is then further adapted to synthetic dataset and miniF2F in section 5 and section 6 respectively.
>
> Please refer to Table 2 for our main result on mathlib and miniF2F.
>
> ## Experiments
> > The experiments are not extensive enough. Most of the experiments are conducted on the miniF2F dataset, which consists of 244 validation and 244 test formalized statements of mathematical problems. However, miniF2F is limited to a small data scale, making the results not solid enough.
>
> Our experiences are conducted on 3 datasets, mathlib, synthetic inequalities, and miniF2F.
>
> In Table 2 we report our main results on 2 datasets, mathlib and miniF2F. We would like to note that the result of mathlib-valid surpasses the state-of-the-art at the time of preprint writing (62.6% vs 48.4% for the same search budget, 75.8% vs 48.4% for full comparison).
>
> For each dataset, we conduct 8-9 expert iterations (c.f. Appendix H in revised version for compute resources required, ~2000 A100 day of compute). We provide ablation results in Table 2 running expert iteration only on *mathlib-train*.
>
> > the paper fails to compare more baselines or search strategies in the experiments.
>
> Our paper compares against PACT [B2], which is the state-of-the-art method on the Lean system at the time of preprint writing. We rerun PACT's setting in our new training objective and new mix dataset for a baseline to compare against. We also provide a baseline with equal computation budget without expert iteration. We follow the previous works [B1, B2, B3, B4] to report pass rate at different expand budgets during search.

---

> > ### Author Response · Authors · 2022-11-11
> > **Response (2/2)**
> >
> > ## Contribution and Novelty
> >
> > > The concept of expert iteration is not novel in the related work
> >
> > We do not intend to claim expert iteration as our novelty. Instead, we would like to stress that our novelty is showing that using expert iteration, at the same computation budget, outperforms sampling only strategy. This is non-trivial as formal math is equipped with a game-like environment which could reject invalid input, making rejection sampling quite efficient and used in previous works [B1, B2].
> >
> > >  the paper fails to discuss the contributions of the proposed method.
> >
> > We would like to note that expert iteration is adopted to the field of neural theorem proving for formal math in this paper through introducing a curated set of curriculum statements. The result of this paper directly leads to solving several difficult olympiads problems (c.f. Appendix J.1 in revised version) and higher pass rate compared to previous methods on both mathlib & miniF2F.
> >
> > Our proposed method points to a potential future that the lack of self-play in the formal mathematics setup can be effectively compensated for by automatically/manually curated sets of formal statements, which are much cheaper to formalize than full proofs and can be further scaled with automated generation. This finding is referred to by subsequent work [C1] for auto-formalization.
> >
> > As another component of contribution, we also contributed a REPL interface of Lean and a set of curriculum statements for the use of the community. This set of statements is also utilized in subsequent work [C2] to further boost the performance on miniF2F.
> >
> > ## Writing and Organization.
> > > the paper misses the reference when mentioning Go on page 2 and misses the reference for Lean in the related work section.
> >
> > We added reference to Go in revised version. We separately introduced Lean in Section 3, Formal Environment, with an introductory paragraph.
> >
> > > The statement "These two differences make a naive application of reinforcement learning to formal mathematics unlikely to succeed." lacks the necessary supporting facts in the paper.
> >
> > Thank you for pointing this out. We’ve changed the wording in the revised version and added supported reference.
> >
> > > Typo.
> >
> > Thank you for pointing out the typos. We will incorporate all suggestions.
> >
> >
> > ### Reference:
> >
> > [A1] Daniel Selsam, Kevin Buzzard, Reid Barton, Percey Liang, Sarah Loss, and Freek Wiedijk. Imo grand challenge. https://imo-grand-challenge.github.io/. 2019.
> >
> > [B1] Stanislas Polu and Ilya Sutskever. Generative language modeling for automated theorem proving. CoRR, abs/2009.03393, 2020.
> >
> > [B2] Jesse Michael Han, Jason Rute, Yuhuai Wu, Edward W. Ayers, Stanislas Polu. Proof Artifact Co-training for Theorem Proving with Language Models. ICLR 2022.
> >
> > [B3] Albert Q. Jiang, Wenda Li, Jesse Michael Han, and Yuhuai Wu. Lisa: Language models of isabelle proofs. 6th Conference on Artificial Intelligence and Theorem Proving, 2021.
> >
> > [B4] Kunhao Zheng, Jesse Michael Han, and Stanislas Polu. minif2f: a cross-system benchmark for formal olympiad-level mathematics. ICLR 2022.
> >
> > [C1] Yuhuai Wu, Albert Qiaochu Jiang, Wenda Li, Markus Norman Rabe, Charles E Staats, Mateja Jamnik, Christian Szegedy. Autoformalization with Large Language Models. NeurIPS 2022.
> >
> > [C2] Guillaume Lample, Marie-Anne Lachaux, Thibaut Lavril, Xavier Martinet, Amaury Hayat, Gabriel Ebner, Aurélien Rodriguez, Timothée Lacroix. HyperTree Proof Search for Neural Theorem Proving. NeurIPS 2022.

---

### Official Review · Reviewer_pG4L · 2022-10-28

**Confidence:** 4
**Correctness:** 4
**Technical Novelty And Significance:** 3
**Empirical Novelty And Significance:** 3
**Recommendation:** 8

**Clarity, Quality, Novelty And Reproducibility:**

The paper is well-written is mostly clear. I've mentioned above what I consider to be new in the paper.

Questions: A few more ablations might be insightful though I recognize some of these could be expensive.

Specifically, in section 5.2, St is taken to be the union of mathlib-train and synth-ineq. What is the performance of the case St = mathlib-train on synthetic inequalities? This is a simple extension of the results in section 4.5, and since synth-ineq provides fine control of complexity it would be interesting to see the results.

Similarly, in section 6.1, what happens if St is the union of mathlib-train and miniF2F-curriculum (no synth-ineq). I suggest this ablation as synth-ineq doesn't seem to be particularly closely connected to miniF2F, and so it would be good to know if it contributes to the improvement in performance

I didn't follow the sentence: "…closed, suggesting a lack of density in our manually formalized set of statement."




**Strength And Weaknesses:**

The main new conceptual element of the present paper, as I understand it, is that bootstrapping also leads to the model becoming able to prove harder theorems when provided with *statements* of theorems of increasing difficulty in the training data. The authors design two curriculums of progressively harder theorems. One is a synthetic dataset based on inequalities and the other comes from mathematics problems.

The use of proof length objective is also a new element over previous work.

The paper provides a few illustrative, if cherry-picked, examples of the model-produced proofs which are shorter than the proofs in the ground truth. I found these interesting. There's a useful discussion of the limitations of the present work.

(The results in the current paper have been improved in subsequent work but I've tried to not take that into account in my review.)

**Summary Of The Paper:**

The paper applies a bootstrapping procedure inspired by AlphaZero to the problem of mathematical theorem proving in a formal language (Lean 3) that's powerful enough to capture most of mathematics. When solving a problem, often mathematicians first generate ideas and then verify if it leads to progress. Somewhat analogous architecture is used here: (1) A large language model (LM) generates "ideas". In this case, it means tactics with appropriate arguments (tactics come from a library of powerful procedures that transform a given goal that needs to be proven to a new, possibly easier to prove, goal(s)). (2) The verification is then done symbolically (in this case, using the Lean Theorem Prover).

Initially, the LM is trained on (among other things) on the Lean mathlib, which is a library of theorems and proofs in Lean. Bootstrapping (also referred to as Expert Iteration in the paper) then consists of generating new proof search trees using the LM, and using this data to further fine-tune the LM. This is repeated up to 9 times.

This general procedure has appeared in previous works as mentioned in the paper. The previous work has demonstrated that bootstrapping leads to better performance in terms of the number of theorems proved. The current paper provides lean-gym which is a useful tool for carrying out the searches and could be useful for others.


**Summary Of The Review:**

The paper shows that expert iteration is capable of learning to prove increasing difficult theorems when provided with a curriculum of statements of increasing difficult theorems.

---

> ### Author Response · Authors · 2022-11-11
> **Response**
>
> Thank you for your review and your time dedicated to this.
>
> > in section 5.2, St is taken to be the union of mathlib-train and synth-ineq. What is the performance of the case St = mathlib-train on synthetic inequalities? This is a simple extension of the results in section 4.5, and since synth-ineq provides fine control of complexity it would be interesting to see the results. in section 6.1, what happens if St is the union of mathlib-train and miniF2F-curriculum (no synth-ineq). I suggest this ablation as synth-ineq doesn't seem to be particularly closely connected to miniF2F, and so it would be good to know if it contributes to the improvement in performance
>
> These are interesting questions, while the computation resources required for these ablations are huge (running through expert iteration requires 2000 A100 day of compute, c.f. Appendix H in revised version). We try to share our thoughts here:
>
> - We did have early-stage results in our internal experiments using only St=*mathlib-train*. Excluding *synth-ineq* from St, on which the new proofs are generated, made our models lack sufficient demonstration (especially for more complex ones) on *synth-ineq* to kick-start the bootstrapping and thus are stuck in low-level synthetic inequalities. In this setting our model did not show signs of curriculum climbing.
>
> - We design *miniF2F-curriculum* to also include a large portion of inequalities problems. The proofs found on *synth-ineq* have needed ingredients (e.g. the demonstration of how to prove AMGM, Cauchy, etc. and how to break down complex inequalities to these elementary inequalities) to close some inequalities statements in *miniF2F-curriculum*, making the new proofs training set potentially larger for next iteration. Excluding *synth-ineq* might result in less new proofs on *miniF2F-curriculum* that are related to inequalities and thus slow down the curriculum climbing effect.
>
> > didn't follow the sentence: "...closed, suggesting a lack of density in our manually formalized set of statement."
>
> The experiment results (100 out of 327 curriculum statements being proved) potentially point us to the fact that our manually formalized statements are not “dense” enough in the spectrum of difficulty to make the model fully climb the curriculum. We therefore think that more statements are needed to fill in the gap and lead to a larger percentage of closed statements.

---

### Decision · Program_Chairs · 2023-01-20

**Decision:**

Accept: notable-top-25%

**Justification For Why Not Higher Score:**

The audience of this paper should include AI researchers who are interested in theorem proving. However, this paper is not written in a way that is super-clear to people who are not immersed in this field, as indicated by the review of reviewer GFD4. I would not be upset if it were to get an oral, but I would be concerned that the oral presentation would also leave most of the ICLR audience unclear about what exactly is going on.


**Justification For Why Not Lower Score:**

This paper makes exciting progress on an AI grand challenge.

**Metareview: Summary, Strengths And Weaknesses:**

This paper advances the state of the art on a major theorem proving benchmark. Theorem proving is a complex AI problem requiring sophisticated approaches. The present paper introduces a technique involving a curriculum of theorems of increasing difficulty. This is a valuable contribution.

The reviewers were satisfied with the clarity of presentation. While much of the current approach uses ideas from previous work, the reviewers found the new idea important and the results impressive and worth disseminating. The reviewers also found lean-gym, a tool introduced in this paper, to be a useful contribution. The only weaknesses discussed were: (a) possibly making the presentation more "user-friendly" for AI researchers interested in the field who are not intimately familiar with the prior work, and (b) further innovating and/or simplifying the procedure. However, leaving these to future work seems like the right decision.

**Note From Pc:**

if the above contains the word "oral" or "spotlight" please see: "oral" presentation means -> notable-top-5% and "spotlight" means -> notable-top-25%. As stated in our emails, we are disassociating presentation type from AC recommendations

**Summary Of Ac-Reviewer Meeting:**

Reviewer GFD4 was able to have their concerns assuaged by the other two reviewers who had a bit more expertise in the area and decided to raise their score.